# Are Geographical Indication Products Fostering Public Goods? Some Evidence from Europe

**Filippo Arfini** [1] , **Elena Cozzi** [1] , **Maria Cecilia Mancini** [1,*] , **Hugo Ferrer-Perez** [2] and **José María Gil** [2]

1   Department of Economics and Management, University of Parma, 43125 Parma, Italy;
    filippo.arfini@unipr.it (F.A.); elena.cozzi@unipr.it (E.C.)
2   CREDA-UPC-IRTA, Parc Mediterrani de la Tecnologia, ESAB Building, C/ Esteve Terrades,
    8, 08860 Castelldefels, Barcelona, Spain; hugo.ferrer@upc.edu (H.F.-P.); chema.gil@upc.edu (J.M.G.)
*   Correspondence: mariacecilia.mancini@unipr.it; Tel.: +39-0521-032383

**Abstract:** Within the framework of multifunctional conceptualisation, the authors have investigated the level of public goods embedded in Agri-food geographical indication products. Moving from the concept of the local Agri-food system, the generation of public goods are observed both on the value chain and on the territory. Three different dimensions of public goods are considered: Cultural heritage issues, socio-economic themes, and natural resources. To pursue this aim, the FAO-SAFA method is adopted. A single index for the three dimensions is computed in order to provide an easy and quick interpretation of the three dimensions. Preliminary empirical evidence on two cases studies suggests different public goods levels embedded in geographical indications, depending on the dimensions analysed. The method proposed aims to be a simple and effective tool to support good practice for policy makers and indicate fields for intervention where indexes show that improvements could be made.

**Keywords:** geographical indications; public goods; sustainability; Parmigiano Reggiano; Ternasco de Aragón

## 1. Introduction

Geographical Indications (GIs) as a Quality Scheme product category were introduced into the market under the Agreements on Trade-Related Aspects of Intellectual Property Rights (TRIPS) of the Uruguay Round (1994). They have been effective in Europe, in a "sui-generis" model, since 1996 under Regulation 2082/1992, amended by the Regulations 510/2006 and 1151/2012. In the European Union (EU), GIs are considered to be an important political tool for several objectives: (i) To guarantee fair competition for farmers and producers of agricultural products and foodstuffs having value-added characteristics and attributes; (ii) to reduce consumers' information asymmetry pertaining to such products; (iii) to foster rural development objectives in rural areas. For all these reasons, GI-policy in the EU needs to be considered not just as a sectoral policy, but as public policy aiming at delivering public goods (PGs) to the whole of European society.

Since 1996, European GIs have been studied under different perspectives. The debate has been intense, as shown by several congresses (Parma 1998; Le Mans, 1999; Parma 2015) and EU research projects (Dolphins; Typic, Siner-GI; Strength2Food), focussing on topics such as effectiveness in fostering the economic performance of producers, the implications for trade on international markets, the implication for rural development, consumers capacity to recognise and trust GI products, consumer willingness to pay (WTP), the capacity to increase the sustainability of the value chains related to those products, and to the related areas of production [1], and, finally, GI capacity to generate PGs.

The significance of the classical theorization of PGs is linked to other features as well as the GI intrinsic feature of being club goods. Classical theorization is highly relevant for the analysis of the range of collateral effects produced by GIs themselves due to the specificity of the joint production function which characterizes the production of foodstuffs and the close links with the territorial and environmental features of the production region. Researchers [2] (p. 148) in fact have noted the need to consider the "collective dimension", along with the "individual dimension", to reach a wider evaluation and knowledge of the effects of GIs. In considering all the positive externalities of GIs in terms of natural resources, animals, and human traditions, it is useful to adopt the notion of PGs.

The concept of PGs first emerged in the mid-20th century [3], when the role of government in providing goods was advocated. The original conceptualization was linked with state intervention in providing goods where the private sector and the market failed to do so. Along with this interpretation, the concept has always included another core feature: Citizen benefit in accessing these goods. In the following decades, the nature of a PG nature was increasingly analysed with a focus on citizens' access. PGs have been classified using different degrees of rivalry and excludability, as shown in Figure 1.

|  |  | Degree of rivalry | |
|  |  | **High** | **Low** |
| | **High** | Private Goods | Club Goods |
| **Degree of Excludability** | **Low** | Common Goods | Public Goods |

**Figure 1.** Typology of goods.

Starting from a common definition of PG as low-rivalry and low excludability, a further definition is "procedural" [4] (p. 47), which makes it possible to consider the production side of the goods on one hand, and the distribution of benefits for the whole society on the other.

Considering agriculture and food production, the framework linking the generation of PGs with GIs was clearly the "multifunctional concept" introduced by the Organisation for Economic Co-operation and Development (OECD) [5]. This framework refers to the economic activities which produce multiple and interconnected results and effects. These results and effects may be "positive or negative, intentional or unintentional, synergetic or conflictive, and valued on the market or not" [6] (p. 27). The main concept supporting the idea of multifunctionality is the ability of food production systems to support the generation of positive and negative externalities within the multiple roles played by agricultural activities: Production of food and fibres, landscape, and environmental preservation, and stimulus for rural areas employment. This concept is useful in the analysis of the links between agricultural policies and PGs related to the generation of commodity and non-commodity goods.

Considering Food Quality Schemes (FQSs), the assumption is that GIs contribute to the generation of PG by positive externalities for the benefit of value chains and rural areas thanks to their positive impacts on natural resources, cultural heritage, and socio-economic spillover effects. The links between PGs and GIs is clearly stated by the Food and Agriculture Organization of the United Nations (FAO) [1] which analyses how GIs are generating positive externalities in term of preserving natural resources, cultural heritage, food security, and employment in lagged regions. Belletti et al. have extended the levels of the GI effects, defining five "publicness profiles": The impacts on environmental and "human" resources and on socio-economic variables, and the effects on social capital, and on secondary businesses linked to the GI [4].

More generally, it is possible to group PG issues affected by agriculture activities (including GI production) into the following categories: Environment, rural development, food safety, and food security, and animal welfare [7]. Within these categories, it is necessary to consider both positive and negative externalities, and their value must be assessed, communicated and paid in order to

avoid market failure, to allow the reproduction of the GI systems, and to promote a more sustainable production and consumption pathway [1].

Those assumptions are validated in the literature by qualitative descriptions of GI impact on those elements that can be considered as PGs [8,9]. Few empirical analyses have been carried out: Raimondi et al. [10] confirmed the benefit of GIs from a macro-economic perspective in terms of local development, focusing the attention on two variables: Sectoral labour productivity and sectoral employment. Observations were made in three European countries over a period of 22 years. Belletti et al. [4] looked at the wine and coffee value chains in order to assess whether GIs can support the supply of PG.

The characteristic of GIs of improving producer income by generating a price premium that internalizes the value of PGs is an important field of research. Studies focusing on this topic can be grouped into studies on economic performance from the producer perspective [11] and studies on the WTP from the consumer perspective [12,13].

From the producer perspective, Barjolle [14] analyses GIs in their supply-chains and managerial features, while Paus and Reviron [15] propose a review of the different research, considering objective and subjective methods as evaluation criteria. Objective assessment methods, which are based on the "historical approach" [15] (p. 16), underline the role of GIs comparing different scenarios on a temporal basis, before and after the adoption of a GI. In other cases, GIs are compared within the supply chains, and the success of individual GIs assessed through an increase in price or volume. In some cases, the whole value chain or territorial system is observed, rather than the individual farm, and benefits with the different cost categories generated by the quality scheme are compared [2]. From these analyses, the main concept emerging is the link between GIs and PGs; a link which consists of quality and reputation. The first aspect is an intrinsic quality attribute generated by the definition of a Code of Practice (CoP) which aims to prevent market failure due to fraud imitation and unfair competition and enhance GI reputation. The legal feature of GIs, i.e., trademark law vs. sui-generis system, and the enforcement rules in different countries are also fields of significant interest widely explored by scholars [16].

Research emphasises that consumer WTP is closely linked to the goods' intrinsic qualities, which are recognized as "credence or trust attributes" [17,18]. Information asymmetry between producers and consumers is the main issue that affects consumer behaviour and WTP. The literature describes different mechanisms that can be adopted to lower information barriers between producers and consumers [19,20] (pp. 58–65). In fact, communication strategies vary in relation to the features of the value chain and to the commercial outlet chosen by producers for their channel. Moreover, although the theory of convention provides a guide, it is argued that "no unifying theory for the sustainability construct exists in communication strategy and consumer behaviour" [20] (p. 60). In sum, research focusing on the consumer perspective is more frequent than research carried out on the production side. Apart from Belletti [21], few studies have investigated the reliability of GIs to generate PGs.

A coherent conceptual framework taking into account the ability of FQSs in general, and of GIs in particular, to link together the production systems with the consumer behaviours and to generate PGs, needs to include the different types of externalities arising from the GI system as well as the stakeholders which are beneficiaries (or damaged) by such externalities. With regard to externalities, it is possible to observe how a GI system is able to produce different types of PGs that can be traced back to cultural, environmental, social, and economic externalities. The level of externality generated depends on the features of the CoP, on the commercial and economic strategies firms adopt, and on the social and environmental features of production and consumption patterns. It is clear that potential users of externalities can be either a member of the value chain (producers and/or consumers) or simple citizens living in the production areas. In the light of this distinction, the externality specification linked to PGs has a different meaning and value according to whether it is analysed within the value chain or the geographical region.

A wide methodological approach should integrate the value chain compendium into a territorial production system. This link is provided by the scholars which conceive the concept of Local Agri-Food System (LAFS) [22]. The LAFS adopts a territorial approach to interpreting the ability of local food systems to generate quality value chains and, especially manage them. LAFS can thus consider the value chain embedded in the territory in its environmental, social, and economic components. The type and the size of externalities related to PGs differ in relation to the features of the value chains, which may lie entirely inside the production region, or may have no boundaries for either supply of raw materials or consumer markets (Figure 2).

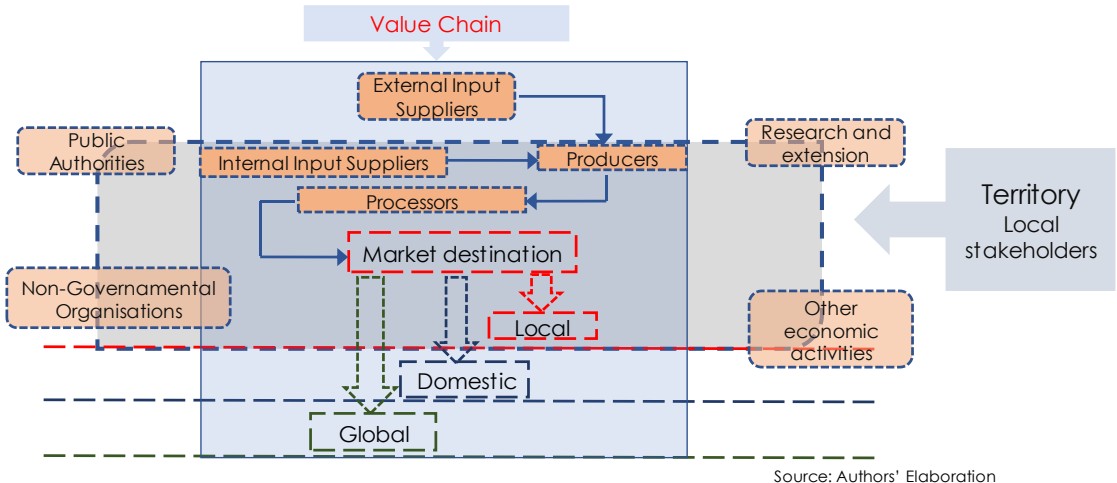

Source: Authors' Elaboration

**Figure 2.** The Local Agri-Food System Approach.

The LAFS approach makes it possible to consider PG generation at the level of the value chain (differentiating Upstream and Processing levels) and at the regional level where this latter can have different size according to the specificity of the production system. In the study of GI systems, it is important to incorporate the territory into the analysis. Muchnik and Sautier in fact, stressed the idea that *"a production structure and the related services are associated through their features and their operation path with a specific territory. The environment, the products, the individual people with their know-how, their institutions, their food habits, and their relation-networks combine in a territory to setup an agri-food organization in a defined spatial area"* [22] (p. 1465) (authors' translation). This formula takes into account all the elements that are relevant for understanding the strategies of LAFS actors concerning production and consumptions as well as assessing the impacts of the different typology of externalities (environmental, cultural/human, and governance) and the related PGs. The concept of LAFS proposed by French scholars enables different analysis: (i) For researchers encompassing the integration of different disciplines (for instance, biotechnical sciences and social sciences), the integration of spatial-temporal scales (sometimes the activities are carried out places which are physically distant and the temporal flow is not always linear), and the different activities [22]; (ii) for economic agents it allows them to develop specific strategies to respect the management of input supply, waste management as well as market strategies; (iii) for policy makers it enables the assessment of sustainability level and generation of PGs.

The aim of the present work is thus to propose a theoretical framework and a method to assess the connection between FQSs (including GIs) and PGs. The methods proposed will be tested on empirical case studies, analysed in the Strength2Food project, using both qualitative and quantitative information. The theoretical context considers both the framework that generates PG indicators and the framework related to quantitative assessment. The final objective is to provide policy makers and stakeholders with an objective set of tools useful to generate information for the management of the GI system, preserving its capacity to generate PGs.

The paper is organized as follows: Section 2 analyses the theoretical framework, and the new methodology is outlined. The methodology is applied to two concrete case-studies, analysed in the results section, and in Section 4, our conclusions are drawn.

## 2. Materials and Methods

*A Theoretical Framework for the Assessment of Public Goods in Food Quality Schemes*

The theoretical framework adopted in the present research follows the logic of the Local Agri-Food System (LAFS) approach which enables us to consider both the chain structure (farming and processing) and the territorial dimension (the area defined by the code of practice). For each level the analysis consists of two steps: (i) The definition of the externalities and the associated public goods (PGs); (ii) the size of the impact of those PGs on the geographical indication (GI) system.

The definition of PG was developed by using specific indicators from different types of externalities. A list of indicators aiming to capture PG generation was developed starting from the approach proposed by the Food and Agriculture Organization of the United Nations (FAO): The sustainability assessment of food and agriculture systems approach (SAFA). The SAFA approach aims at describing the economic, social, governance-related and environmental impacts of agricultural and food systems, with a list of over 100 indicators computed on a self-assessment basis. SAFA isolates 21 themes and 58 sub-themes covering the four above-mentioned dimensions [23].

As in the SAFA philosophy, in our project, PGs were identified and defined according to three classes of externality: Environmental, socio-economic, and cultural. These classes were chosen because they are closely related to GI features. Indeed, as noted above and in line with the application of the LAFS concept, GIs products have a close link with the terroir. Furthermore, they result from intangible paths involving, for example, particular knowledge inherited from history or particular social features. These values represent key GI intrinsic attributes so that it is important to observe the management of natural resources, landscape maintenance, the state of animal welfare and attitudes towards cultural heritage. Social involvement, governance actions, and the generation of positive social relations and social networks also play a key role in shaping the social dimension.

Table 1 summarises the aspects described above and reports the set of indicators used in the analysis.

This analysis was developed in the S2F project framework, which assesses the impact of FQSs on sustainability and PGs. The amount of input and output related to the production of several FQSs around the world was assessed [24]. A rigorous methodological approach was defined with the aim of assessing the level of sustainability. Specifically, economic (Ec), environmental (En), and social (So) indicators were computed on the basis of primary and secondary data collected in both field and desk analyses. Close co-operation among researchers enabled an accurate interpretation for each technical indicator examined in the analysis.

The objective of this approach is to define benchmarks that can be updated every year in order to provide useful information to stakeholders to allow them to manage the value chain and the territory, maintaining the desired PG level associated with the specific GI system.

In more detail, the indicators computed to assess the sustainability levels in the S2F Project which were selected to represent PG dimensions are:

1. The carbon footprint/product (En1_a) and the carbon footprint/area (En1_b);
2. The water footprint (respectively, green water: En3_a, grey water: En3_b, and blue water: En3_c)

   for the environmental dimension;

3. The labour-to-production ratio (So1_a) and the profit-to-labour ratio (So1_b);
4. Educational attainment (So3_a);

   the generational change (So5_a) in the domain of cultural heritage;

5.　　Bargain power distribution (So2);
6.　　The local multiplier (Ec2) for spillover socio-economic effects.

**Table 1.** List of qualitative and quantitative indicators.

| Class of Public Goods | Indicator | Code | Typology |
|---|---|---|---|
| Indicators concerning Cultural Heritage Preservation | Communication activities | CH_1 | Qualitative |
| | Value chain foreigner workers attraction | CH_2 | Qualitative |
| | Educational attainment | CH_3 | Quantitative (So3_a) |
| | Support touristic events | CH_4 | Qualitative |
| | Generational Change | CH_5 | Quantitative (So5_a) |
| | Labour-to-production ratio | CH_6 | Quantitative (So1_a) |
| | Educational Farm Activities | CH_7 | Qualitative |
| | Professional training on the FQS | CH_8 | Qualitative |
| | Profit-to-labour ratio | CH_9 | Quantitative (So1_b) |
| | PGs definition into CoP | CH_10 | Qualitative |
| Indicators concerning Socio-Economic spillover effects | Participation to farmer unions | SE_1 | Qualitative |
| | Participation to board association | SE_2 | Qualitative |
| | Participation to technical association | SE_3 | Qualitative |
| | Intensity of network relationship | SE_4 | Qualitative |
| | Relevance of cooperation system | SE_5 | Qualitative |
| | Bargain power distribution | SE_6 | Quantitative (So2) |
| | Governance actions | SE_7 | Qualitative |
| | Economic spillover effect | SE_8 | Quantitative (Ec2) |
| | PGs definition into CoP | SE_9 | Qualitative |
| Indicators concerning use of Natural Resources | Animal welfare definition into CoP | NR_1 | Qualitative |
| | Blue water | NR_2 | Quantitative (En3_c) |
| | Carbon footprint per Ha | NR_3 | Quantitative (En1_b) |
| | Carbon footprint per unit of product | NR_4 | Quantitative (En1_a) |
| | Green water | NR_5 | Quantitative (En3_a) |
| | Grey water | NR_6 | Quantitative (En3_b) |
| | PGs definition into CoP | NR_7 | Qualitative |

These indicators are considered proxies in the fields they represent and have a meaning for the value chain and for the territory. Only the local multiplier can be considered a "pure" territorial indicator.

Environmental indicators capture the inputs and outputs which impact on the natural resources and, indirectly, on the capacity for contributing to the reproduction of the local environment, including the landscape.

Social indicators describe the impact on the capacity for contributing to maintaining a proper social structure and cultural heritage. The indicator So1_a was included in the analysis since it represents the cultural dimension of a GI system: PDO and PGI comprise in themselves a high degree of expertise and know-how, which is preserved thanks to human capital. For this reason, a high percentage of work units is considered as a positive contribution to the preservation of know-how and traditions in a local area. The profit-to-labour ratio (So1_b) and the generational change (So5_a) are used for the same reason. Generational change (So5_a) reflects, on one hand, the preservation of traditional production methods, but on the other, it contributes to the viability of rural areas, and, for this reason,

it could also be subject to an extended interpretation. Educational attainment (So3_a) was included because it fosters the creation of social capital, which, in turn, affects heritage preservation.

The economic effect is captured by the bargain power distribution (So2) and local multiplier (Ec2). The distribution of bargaining power represents the ability of the value chain to bargain along the value chain [24], while Ec2 records the flows of money within the local economy generated by €1 income earned from GI production. The computation of Ec2 takes into account three rounds of spending: The amounts of spending retained within the local area is measured in each round. This indicator perfectly reflects the territorial impact and the ability of the GI to generate spillover economic effects in favour of inhabitants of the production area.

Along with the quantitative indicators, qualitative ones were also introduced to capture other GI production system "complex" features which might contribute to PG generation. These features are:

(i) The contribution of FQSs to the non-farm rural economy in terms of auxiliary services. They are conceived as a wide range of activities covering services directly related to the production system, such as collateral consulting services (e.g., chemical labs for milk analysis, extension services, or administrative offices for farmers) or manufacturing activities (e.g., packaging and marketing services), as well as collateral activities (e.g., in the tourism or artisan sectors). In this way it is possible to assess whether the FQS product is considered a fly-wheel for the local area where its production is based;

(ii) The contribution of different governance models to ensure the valorisation of producers' know-how and local resources. Governance is a complex feature with several implications which are not easy to assess [25]. In this case, governance is considered the ability of the system to manage quality and network relationships with the aim of improving market efficiency and social cohesion;

(iii) Social cohesion as a way to booster social capital and social networks. The presence of producer or inter-professional Organisations could in fact influence not only economic performance but particularly global sustainability at a local level, thanks to the representativeness of the interests of all the actors involved.

Information sources suitable for assessing the qualitative indicators were the CoP, interviews with local experts and other grey materials, such as reports, surveys, local web-sites etc. The assessment of each indicator was made on a Likert scale from 1 to 7, where 1 corresponds to the lowest, and 7 to the highest externality.

In the methodology, assessing the dimension of each indicator was an important decisional phase. Only indicators like Carbon Dioxide ($CO_2$), and water ($H_2O$), etc. can be measured by quantitative methods (i.e., kg/ha, litre/ton of output), whereas others have qualitative features. Quantitative assessment is thus not a simple process. Furthermore, each indicator shows a specific meaning expressed on a Likert scale. In order to compare the effects, indicators need to be homogenized on a consistent scale which permits comparison to be made. For this reason, all the quantitative and qualitative indexes were normalized on a scale from 0 to 1, where 0 represents the lowest level (i.e., the lowest contribution to the generation of PGs) and 1 the highest. The normalisation was made in order to obtain comparable indexes (unit less indexes), on one hand, and to summarise them in aggregated indexes, on the other hand. To pursue the first aim, the indicators were simplified and grouped into cultural heritage, socio-economic spill over, and use of natural resources.

The Normalisation of these indexes was another critical step in the research since the definition of the minimum and the maximum level influences the final data. In this phase, the minimum and maximum values for the quantitative indicators were identified from those appearing in the literature. Finally, all the indicators were set on a 0–1 scale, where 0 represents the lowest score or the lowest contribution to the generation of PGs. In this way, it was possible to observe the related indicators through a further step: The calculation of a single index for each PG dimension: (i) Cultural heritage preservation; (ii) socio-economic spill-over; (iii) use of natural resources.

The PGs indexes are multi-dimensional and are intended to describe a complex system of different phenomena captured by single indexes. The problem (and the solution) is similar to the one adopted by the United Nations Development Program (UNDP) in computing the Human Development Index (HDI), which combines the dimension of a long and healthy life with the access to knowledge, and a decent standard of living. In the present study, the challenge was how to treat and how to calculate a "higher" PG index which aggregates single indicators representing the different dimensions.

Among the several methods of weighting and aggregating indicators to be chosen according to the purposes, the scales, and the perspective adopted [26], in this research the method adopted was aggregation through a geometric mean. This is in line with the purpose of assessing the state of a particular production, as pointed out by Gan et al. [26], although we also rely on a strong sustainability perspective (Markulev and Long, 2013 cited in [26] p. 492). The choice of relying on a strong perspective, rather than on the weak one, reflects the idea that all dimensions contribute equally to PG generation and locates the study in the research line, which takes into account other dimensions besides the purely economic one [27].

Consequently, no weighting procedures were implemented among indexes and, to avoid compensability among the dimensions, only a geometric aggregation method was utilised. In fact, using a multiplicative function instead of an additive one, the indicators are not compensated. We thus proceeded by computing one aggregated index per category and then, following the same method, one general PG aggregated index.

## 3. Results

The assessment of the degree of incorporation of public goods (PGs) within geographical indications (GIs) was conducted on two case studies, also included in the S2F Project: Protected Designation of Origin (PDO) Parmigiano Reggiano and Protected Geographical Indication (PGI) Ternasco de Aragón. These two case studies represent different production systems, different links with the environmental features of their production territories, and diverse commercial destinations. The aim of the present research is to evaluate the ability of the proposed methodology to assess the different level of externalities linked to PGs that those GIs can generate.

### 3.1. Parmigiano Reggiano Local Agri-Food System

Parmigiano Reggiano is one of the best known PDO Italian cheeses in the world. Its quality depends on a strict Code of Practice (CoP) which regulates milk production and its transformation into cheese in a defined production area (five provinces in Emilia-Romagna and Lombardy regions) as well as the ripening system and the use of logos in commercial activities. Since 1964, the Consortium (Parmigiano Reggiano Cheese Consortium) has managed activities related to the Denomination of Origin and, since 1992, as manager of the PDO. The Consortium's tasks are: The defence and protection of the Designation of Origin, and the facilitation of trade and consumption by promoting initiatives aimed at safeguarding the typicality and unique features of the product.

In the Parmigiano Reggiano system, natural factors play a central role in typifying the final product. The protection and careful management of the natural resources thus represent an important phase enabling the survival of the uniqueness of the product. For this reason, alfalfa still today is a substantial proportion of the diet of the animals. The cultivation and the use of this forage guarantees a good level of animal welfare and impacts positively on natural resources as well as landscape maintenance. Cultural heritage plays a key role in Parmigiano Reggiano which has a historical background dating back to the Middle Ages. In fact, the know-how and the features relating to tradition are closely described in the CoP and have contributed to maintaining a high level of reputation down the centuries.

The Parmigiano Reggiano system is based on a complex network which includes private and public actors closely connected to each other within the territory [28]. On average, three million wheels of cheese are produced per year, by 393 dairies (60% cooperatives) which process the milk

from 300,000 farmers located inside the designated geographical area. The cheese, aged in a proper storage environment, is sold after 24 months at a premium price compared to other Italian PDO chesses. Nevertheless, despite its economic profitability, the system is suffering from a fall in the numbers of farmers and dairies especially in mountain areas, and the concentration of production into bigger and more economically efficient farms and dairies, leading to a significant reduction in social capital.

### 3.2. Ternasco de Aragón Local Agri-Food System

Ternasco de Aragón is a PGI fresh lamb meat produced in the region of Aragón (Spain). It is one of the most typical meats of the Aragonese cuisine. The *ternasco* is slaughtered at the age of 70 to 100 days, with a live weight of 10 to 13 kg. In Aragón, *ternasco* meat is a common food preparation and is part of traditional recipes of this region. *Ternasco* meat is especially tasty and tender.

Sheep farming is very important in the Aragón region due to its economic, social and environmental impacts. It is found all over the territory, in both non-irrigated and irrigated areas, in plains and mountain areas. The production of *ternasco* is a key factor for social groups settled in harsh and remote areas. The pedo-climatic peculiarities of the Aragonese territory have favoured the development of an important sheep subsector. Several types of sheep can be found with a number of particular qualities, which make them clearly differentiable from the other sheep commonly slaughtered in Spain.

The CoP of the PGI Ternasco de Aragón includes five sheep breeds: *rasa aragonesa, ojinegra de Teruel, maellana, ansotana*, and *roya bilbilitana*, which are considered as Aragón native breeds. The traditional lamb breeding system is characterised by a grazing phase and a stabling phase. Grazing takes place in mountainous areas, which provide the taste of the meat, as described in the CoP by the *Consejo Regulador de la IGP Ternasco de Aragón* (CRTA) [29]. For the production of *Ternasco*, the CoP plays a central role, since it forbids the use of substances that may interfere with the normal rhythm of growth and development of the animal. Furthermore, it specifies that the animals be transported in suitable vehicles so that they do not suffer any alteration or discomfort that could affect their health or physical integrity.

Ternasco de Aragón has a Regulatory Council (CRTA), which is responsible for guaranteeing the lamb quality. In addition, its other functions are the drafting of reports, supervision of compliance with local, national and European regulations, and the certification of each operator involved along the supply chain.

The fundamental requirements to guarantee the origin of the product impose that meat comes exclusively from the authorized breeds from registered farms, located in the production area, and the final product is subject to the analyses specified to be able to guarantee its quality.

The production system of the Ternasco de Aragón is characterized by the presence of large cooperative groups specialised in the processing of meat from lamb, providing an advantage for the commercialization of the meat and guaranteeing consumers on the intrinsic and extrinsic quality of the meat. Distribution of Ternasco de Aragón is basically carried out by three enterprises. Two of them are cooperatives, which means that sheep farmers participate in almost the whole chain.

The Consortium of PGI Ternasco is very active in supporting the development of Ternasco of Aragón market, facilitating contact between customers and distributors. The consortium also promotes the consumption in local restaurants, in the out-of-home sector, through the organization of events and festival to promote Ternasco of Aragón consumption. The Consortium also promotes tourism through the local network "RUT.A Ternasco de Aragón" which includes producers, restaurants, places of interest in Aragón, museums, and sites of natural beauty.

### 4. Discussion

The dimension of each public good (PG)-index was assessed at the upstream and downstream levels of the value chain within the two case-studies areas. The evaluation has been carried out focusing both on the value chain and on the territory where the production system is set, in line with the Local Agri-Food System concept. In fact, because of the intrinsic features of the PGs themselves, the assessment has to also take into account the impacts on the territory where the production is

based. Reflecting the approach described above, the assessments are shown in Figures 3 and 4. The objective was to portray by disaggregated indicators all the variables corresponding each PG in their three dimensions.

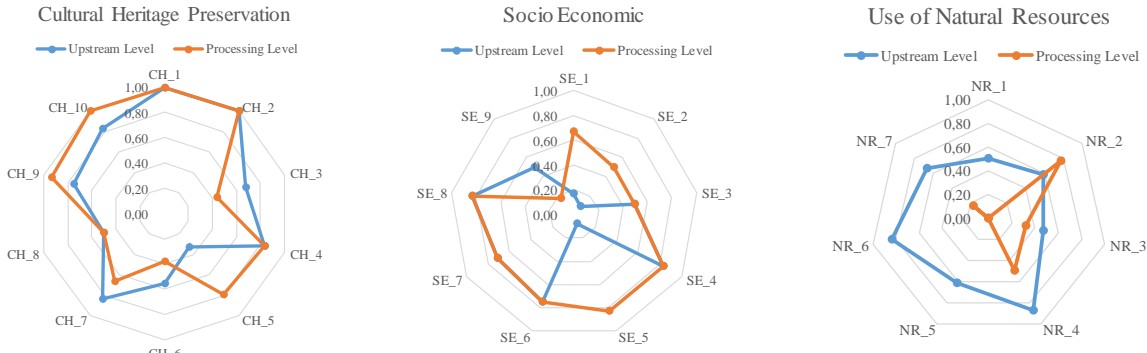

**Figure 3.** Parmigiano Reggiano: disaggregated indicators contributing to public goods (PG) generation.

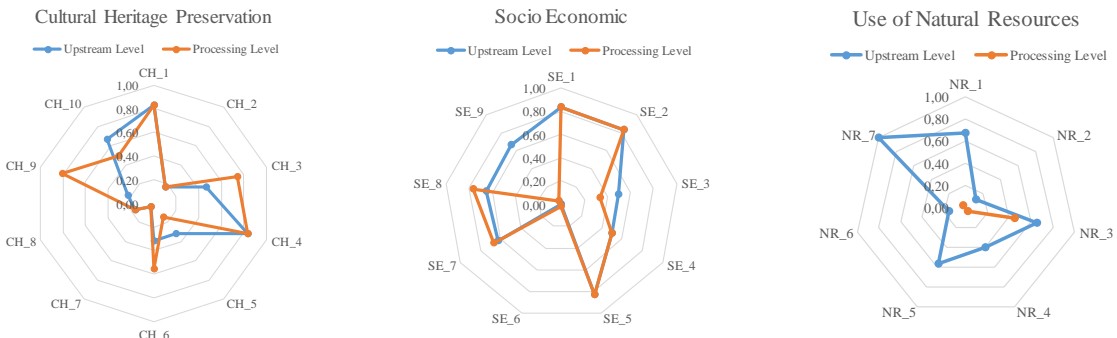

**Figure 4.** Ternasco of Aragón disaggregated indicators contributing to PG generation.

In the case of Parmigiano Reggiano PDO (Figure 3), the use of natural resources, especially at the upstream level, generate positive externalities mainly due to the production of alfa-alfa, which contribute to mitigate the carbon foot print (NR_3, NR_4) and to reduce water consumption. The weight of animal welfare (NR_1) is good but can be improved. On the contrary, at the processing level, the contribution to environmental PGs present a good performance for the blue water (NR_2), while the milk collecting system and the long ageing of cheese lower the carbon emission efficiency. The disaggregated indicators show a high positive contribution to PG related to the use of the natural resource for the upstream level compared to the values of the processing stage. The PG generation is well set and protected in the CoP only at the upstream level (NR_7 is equal to 0, 67) since the techniques to feed cows and the forage origin are strictly codified.

Considering the PG related to the contribution to cultural heritage the disaggregated indicators show a good performance for both the upstream and processing levels. This result reflects mainly two indicators: The outstanding promotion activities carried out by the Consortium (CH_1) and the significant value chain attraction for immigrant workers (CH_2), while a critical aspect is related to the professional training (SH_8) which negatively impact on the PG generation.

Finally, referring to the PG related to the spill over socio-economic effects, as reported in Figure 3, disaggregated indicators clearly show that the social and networking components play a relevant role both at the upstream and processing level; actually, this latter has a major role in most of the variables that are considered. It must be stressed as the contribution to the local economy, catch by the local multiplier index (SE_8), is significant and generates a relevant economic benefit to the whole territory.

Moving to the *Ternasco de Aragón* even if the CoP sets precise rules and specifications regarding the traditional breeding system and genetic breeds, the only indexes related to the animal welfare

and carbon footprint per ha (NR_1, NR_3) generate an acceptable use of natural resources. Moreover, the PG associated to the cultural heritage preservation present strong elements related to the support of touristic events (CH_4), communication activities (CH_1) and to the generation of a high profit to labour ratio (CH_9). On the contrary, the indexes related to the attraction of foreign workers (CH_2) and professional training (CH_8) poorly contribute to cultural heritage preservation. Finally, the socio-economic indexes representing the collective dimension of this production system play a relevant role both at farm and processing levels. This latter contributes to generating a relevant spill over effect for the whole territory.

If disaggregated indexes contribute to set which variable plays a positive or negative role for the generation of PGs, consumers can be solicited by a more concise indicator that quantifies the overall capacity of the production system to generate PGs. To this aim an aggregated index that group together the disaggregated ones may help to communicate the overall benefits to the chain, to the region and to the consumers (Figure 5). Comparing the two case studies, it appears that the Parmigiano Reggiano generates a higher level of public goods and presents a high capacity to preserve cultural heritage and natural systems. On the contrary, the research highlights that in the Ternasco de Aragón case study, even if it is a PGI product, it economically contributes to maintaining the vital upstream level of the LASF (especially in remote areas), while the cultural and environmental aspects play a minor role.

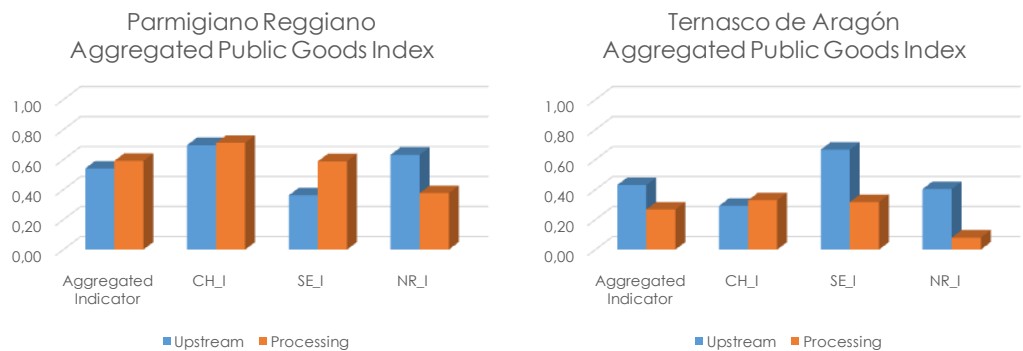

**Figure 5.** Aggregated Public Good Index.

## 5. Conclusions

The methodology proposed to assess the level of externalities associated with public goods (PGs) can be likened to the "dashboard" approach used in a corporate social responsibility analysis. The aim is to describe the "social implications" of corporate activities for stakeholders and consumers, with the aim of identifying useful behaviour for society's aims and thus contribute to a better quality of life. Dashboards can be a useful tool to monitor the evolution of the generation of PG by food quality schemes across time and to indicate appropriate policies for policy makers and stakeholders.

Another important result of the present research is to localise the origins of positive and negative externalities and the consequent public good. We have demonstrated as PGs are generated not only at the value chain level but also at the territorial level; actually the two dimensions are embedded and link together not only the "spatial" dimension of the production systems but also the productive, strategic and commercial ones, linking consumers' expectations to the perception of the GI quality product. There is a strong "cause-effect" relation between the policy maker decisions and impacts at the territorial level, and such decisions influence the level of PGs, the quality of the product and the quality of life in the production areas which are often rural ones. Considering this output, the authors strongly believe that the LASF approach includes most of the elements which characterise geographical indications and the generation of PGs, providing an effective analytical tool.

It is true that an objective assessment of PG indexes requires a great effort in the collection and processing of data as well as close collaboration between local stakeholders and GI managers. Data collection can be a further administrative burden. However, the results of the analyses on the

level of PGs confirm how GIs can effectively contribute to the development of rural viability and vitality enhancing and guaranteeing the level of intrinsic attributes incorporated in GI products.

Finally, an aggregated index makes it possible to compare the impact of PGs, providing fruitful information to stakeholders and policy makers with the aim of adopting a strategy to reproduce the GI system over time and contribute to the sustainability of production areas.

**Author Contributions:** The authors equally contributed to the current research paper. F.A. wrote the Introduction section, M.C.M. the Materials and Methods, E.C. the Parmigiano Reggiano Local Agri-Food System, H.F.-P. the Ternasco de Aragón Local Agri-Food System, J.M.G. the Discussion. All the authors share and wrote the Conclusions.

**Funding:** This research was funded by the European Union's Horizon 2020 research and innovation programme under grant agreement No 678024 for the Strength to Food Project.

**Conflicts of Interest:** The authors declare no conflict of interest.

## Abbreviations

The following abbreviations are used in this manuscript:

| | |
|---|---|
| CoP | Code of practice |
| FAO | Food and Agriculture Organization of the United Nations |
| FQS | Food Quality Scheme |
| GI | Geographical indication |
| LAFS | Local agri-food system |
| OECD | Organisation for Economic Co-operation and Development |
| PDO | Protected designation of origin |
| PG | Public good |
| PGI | Protected geographical indication |
| SAFA | Sustainability assessment of food and agriculture systems indicators |
| TRIPS | Trade related aspects of intellectual property rights |
| WTP | Willingness to pay |

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
