# Peer review of "Are Geographical Indication Products Fostering Public Goods? Some Evidence from Europe"

_sustainability, doi:10.3390/su11010272_

Round 1

Reviewer 1 Report

My in line comments

Line 20, are abbreviations permitted in the abstract, authors to check?

Line72, In general, I really like the introduction for balancing the requirement for GI in terms of provenance and their potential use in regulating trade with TRIPs, this is well put, well structured and good to read

Line 82, I hope this Ms helps to define 'environment', this would be most useful to the sector

Line 134, The authors may consider their heavier use of abbreviations in the MS and actually spell out general terms like Food Quality Systems, this is a suggestion and possibly minor amend should the authors wish to do it, my only critique of what is a really good approach so far.

Line 155, Figure 2, this is very useful to policy and NGO's, how it can be utilised by all supply chain partners is of most impact e.g. SMEs?

Line 183, I see how the authors are defining 'environment' something to consider is complexity; are 21 themes and 58 sub themes actually practical for food supply chain partners to utilise?

Line 197, Table 1, Very good and applicable, I would say that this is step-changing and insightful

Line 412, fig 3, Again, my only criticism is the ease of reading and this is a minor criticism the authors can consider. The use of abbreviation for indications rather than the terms themselves make it more difficult to follow. I like the graphics, presentation and approach and I believe it to be a great relevance to the food sector.

Line 431, Figure 5, this is good, the synthetic indicator might be termed the 'sum' in the food sector? Synthetic can mean something else to consumers, it may be something the authors wish to consider

Excellent paper, well presented, innovative work that is relevant to many partners across food supply chains, not only policy makers. In this light a strong definition of 'public goods' would be useful at the beginning e.g. are we really talking about 'consumer goods/FMCG/foods'? I believe that making this very clear will engage more readers.

I do think the authors really should consider less abbreviations and the use of whole words where and as possible. I know this is difficult. The reason for this is, papers and ideas contain in papers such as this have primary importance to supply chain partners but they will not read them in detail because of too many abbreviations and the text will not flow easily to the reader. The authors should consider this because this paper and study is of high importance outside of the policy and NGO sector. I enjoyed reading it and it was very well presented.

Minor revisions are put forward as suggestions to be considered by the authors, they do not necessary have to change the current MS of they feel they do not need to, it is good enough for publication directly 'as is'

Author Response

The authors thank the referee for the useful comments provided. We have considered them properly. In the following section the details of our change.

Regarding the use of abbreviations: we have added a list of abbreviations at the end of the paper and we have ‘recapped’ each abbreviation at the beginning of each paragraph (and in the titles), in order to let the reading easier and more fluent. We removed the abbreviations from the abstract as well, following the same purpose of clarity and simplicity.

Concerning the 5th comment: We consider Figure 2. useful for all the subjects involved in a GI production, not only for public bodies or institutional organizations. The SME could place themselves in or outside the territory and evaluate their relationships with the other bodies/authorities etc. It could be useful to understand where and how they are related to the other subjects of the value chain, as well as their impacts on the territory (in terms of raw materials supply, market influence and direct impacts on the local area).

Concerning the 6th point: If we have right understood the comment, the SAFA approach has been adopted several times by different value chains and for this reason it has a practical and direct use by the partners in a value chain.

Concerning the 9th point: we have renamed the single indicator. We decided to identify it as “aggregated indicator”, instead of “synthetic indicator”, in order to avoid misunderstanding and to suggest the logical operation behind it (i.e. the aggregation of different dimensions).

Reviewer 2 Report

Discussion of the results is required, it is now missed. Conclusion should be revised such as to reflect not only the direct findings but also the conclusions from the discussion section.

Methodology section is extremely long compared to other parts of the manuscript. Many parts now presented in methodology may be removed or merged with introduction or literature review

Author Response

The authors thank the referee for the useful comments provided. We have considered them properly. In the following section the details of our review.

- Discussion of the results is required, it is now missed. Conclusion should be revised such as to reflect not only the direct findings but also the conclusions from the discussion section.

The discussion was totally re-written linking the results to the general debate. At the same time the conclusion keeps in consideration the main findings of the previous section

- Methodology section is extremely long compared to other parts of the manuscript. Many parts now presented in methodology may be removed or merged with introduction or literature review

The methodology section has been re-organised by moving some issues in the first paragraph. In this way we have followed the suggestion to equilibrate all the Sections.

Reviewer 3 Report

Interesting topic with interesting approach with well esatblished design.

However some minor (formal) adjustments in the text is required, e.g. in line 235 you left unnecessary question marks, in line 266 H2O is misspelled (zero instead of O) etc.

Author Response

The authors thank the referee for the useful comments provided. We have considered them properly. In the following section the details of our review.

We have checked the spelling and made all the amendments. We hope the paper looks like more clear and organized with less grammar and orthographic mistakes.

Round 2

Reviewer 2 Report

The manuscript has been improved substantially, most of my recommendations have been addressed properly.